# Identifying Coping Strategies Used by Transgender Individuals in Response to Stressors during and after Gender-Affirming Treatments—An Explorative Study

**DOI:** 10.3390/healthcare11010089

**Published:** 2022-12-28

**Authors:** Anna O. J. Oorthuys, Maeghan Ross, Baudewijntje P. C. Kreukels, Margriet G. Mullender, Tim C. van de Grift

**Affiliations:** 1Department of Plastic, Reconstructive and Hand Surgery, Amsterdam University Medical Centers, Location VUmc, 1081 HV Amsterdam, The Netherlands; 2Amsterdam Public Health Institute, 1081 BT Amsterdam, The Netherlands; 3Department of Medical Psychology, Center of Expertise on Gender Dysphoria, Amsterdam University Medical Centers, Location VUmc, 1081 HV Amsterdam, The Netherlands

**Keywords:** transgender persons, adaptation, gender-affirming treatments, coping strategies, mental health

## Abstract

Background: Gender-affirming treatments are reported to improve mental health significantly. However, a substantial number of transgender individuals report a relapse in, or persistence of, mental health problems following gender-affirming treatments. This is due to multiple stressors occurring during this period, and in general as a consequence of widespread stigma and minority stress. Aim: The aim of this pilot study was to identify different coping strategies that transgender individuals use in response to stressors prior to and following gender-affirming treatments, as mediator of mental health. Methods: Qualitative interviews were conducted to better understand the treatment outcomes and healthcare experiences of Dutch transgender individuals who had received gender-affirming treatments. Nineteen participants were included, of which 12 identified as (transgender) male, six as (transgender) female and one as transgender. Outcomes: Inductive coding and theory-informed thematic analysis were used to assess stressors (n_codes_ = 335) and coping strategies (n_codes_ = 869). Results: Four stressor domains were identified, including lack of support system, stressors related to transition, and physical and psychosocial stressors post-transition. We identified six adaptive coping strategies of which acceptance, help seeking and adaptive cognitions concerning gender and transition were reported most frequently. Of the seven maladaptive strategies that we identified, social isolation and maladaptive cognitions concerning gender and transition were the most-reported maladaptive coping strategies Clinical implications: The results indicated that transgender individuals may experience significant stress, both transgender-specific and non-specific, prior to and following gender-affirming treatments and, as a result, use many coping strategies to adapt. Increased awareness of stressors and (mal)adaptive coping strategies may help to improve mental healthcare and overall support for transgender individuals. Strengths and Limitations: This is the first (pilot) study to provide insight into the range of stressors that transgender individuals experience during and after gender-affirming treatments, as well as the variety of coping strategies that are used to adapt. However, since this was a pilot study assumptions and generalizations of the evidence should be made cautiously. Conclusion: Results of this pilot study showed that transgender individuals may undergo significant stress during and after gender-affirming medical treatment related to the treatments and the social experiences that occur during this period, and as a result, use a range of coping strategies to adapt to the stress.

## 1. Introduction

Gender dysphoria refers to the distress that a transgender individual may feel regarding a discrepancy between an individual’s experienced gender identity (male/female/non-binary, etc.) and their sex assigned at birth (male/female) [1]. With that being said, not all transgender individuals experience gender dysphoria or seek to medically transition. However, for those individuals who do seek medical transition, gender-affirming treatments aim to alleviate feelings of gender dysphoria and the resulting ramifications that said dysphoria may have on the individual’s mental health [2,3]. Many studies have observed elevated rates of anxiety, depression, trauma, and substance abuse in transgender individuals’ pre-transition, as well as an increased risk of suicidality [4,5,6,7,8].

Gender-affirming treatments such as hormone replacement therapy are reported to alleviate feelings of gender dysphoria, improve mental health, and improve quality of life significantly [4,9,10,11]. However, multiple stressors can often occur throughout an individual’s gender-affirming medical treatment, and as a possible result, a significant number of transgender individuals have reported relapse or persistence of psychiatric problems [9,12,13].

At present, gender-affirming medical treatment is mainly organized around aligning physique and identity; however, no standard psychotherapy/counseling is provided on integrating the two following gender-affirming medical treatment [2]. Generally, once the individual has completed their medical transition, little information is provided on how to cope with certain stressors, little attention is given to the individual’s integration of the physical changes post transition and the resulting relationship with their identity [14]. This lack of aftercare guidance could possibly result in persisting distress, dissatisfaction, and/or difficulties with adapting to life post medical transition [14,15]. A recent study investigated the needs of transgender individuals after having had gender-affirming treatment and found that participants expressed a need for additional mental health aftercare [14]. Altogether, the persistent stressors and limited aftercare that transgender individuals receive may contribute to the increased risk of mental health problems, or relapse therein [9,12,16].

Stress is defined as the response to a situation or any type of change in which psychological, emotional, or physical strain is caused in an individual [17,18]. Numerous studies have reported the increased number of stressors transgender individuals experience (minority stress, widespread stigma), and Meyer et al. have specifically defined two categories under which these stressors tend to fall: distal (e.g., discrimination) and proximal (e.g., internalized transphobia) [19,20]. That said, gender-affirming medical treatment can be a lengthy trajectory encompassing multiple treatments, as well as numerous additional possible stressors (e.g., complications, disappointing results, persisting socio-economical stress) [2]. An individual’s reaction to stress is dependent upon multiple factors, amongst which are the coping strategies that function as a moderator to prevent or exacerbate psychological suffering [17]. Coping strategies can be thoughts or actions that are used to manage or attempt to stabilize stressful situations [21]. Within individuals experiencing various types of psychopathology, adaptive coping strategies have been linked to improved clinical recovery and long-standing positive effects on psychological functioning [22,23]. Research in cisgender populations has shown that coping strategies can be adaptive or maladaptive and may include cognitive, behavioral or emotional efforts that are used to adapt to stressful events [17]. Cognitive coping relies on mental activity to manage stressful situations, whereas behavioral coping uses actions to adapt to stress [24]. Maladaptive coping strategies are often unconscious intentional (cognitive) actions used by the individual to (temporarily) alleviate feelings of stress that tend to be effective in the short-term; they are likely to have a negative impact on the individual’s (mental) health if continued in the long term [25]. The identification of coping strategies as (mal)adaptive depends on the individual, the nature of the stressor, culture, and many other factors. Literature indicates that it is not always possible to classify single strategies as maladaptive or adaptive because often multiple strategies are combined and what are considered effective coping strategies for one individual may be ineffective for another [25,26]. Moreover, an assessment and classification of the mal(adaptive) coping strategies used by transgender individuals is not yet supported empirically.

Few studies have focused on the overall coping strategies that transgender individuals use in everyday life, let alone the strategies that are used in order to adapt and cope with the changes both during and following gender-affirming treatments [27]. A more comprehensive understanding of the experienced stressors and (mal)adaptive coping strategies of transgender individuals during medical transition may assist in the development of specialized (after)care and further support regarding the individual’s ability to adapt and integrate. In this regard, an understanding of both population-specific and non-specific stressors and strategies is of importance.

At present, while transgender individuals experience increased levels of stressors and report higher incidence of mental health problems, the possible role of coping strategies remains unknown. This qualitative study aims to identify different (adaptive and maladaptive) coping strategies used by transgender individuals during and following gender-affirming treatments in response to the individual’s experienced stressors. Additionally, the extent to which the reported strategies are specific to transgender individuals and the specific stressors that they experience, as well as possible approaches to (after)care, will be explored.

## 2. Methods

### 2.1. Study Setup

The present qualitative study reports on semi-structured interviews conducted with transgender individuals who were seeking gender-affirming treatments in the Netherlands. Data were collected as part of the GENDER-Q study [28], and the current study reports on secondary analyses of the Dutch study participants. In 2018, the GENDER-Q study was initiated by researchers in four countries who sought to develop a patient-reported outcome measure (PROM) for adolescent and adult transgender individuals receiving gender-affirming treatments [28]. In the initial part of the study, researchers conducted interviews on a range of topics related to the outcomes of gender-affirming treatments. For the present study, ethics approval was granted by the Ethical Review Board of the [blinded for review] (no.2017.617). Written and oral informed consent was provided by all study participants.

### 2.2. Participants and Recruitment

Study participants included Dutch individuals aged 16 years and older who were recruited for the GENDER-Q study at the [blinded for review]. The complete study design of the GENDER-Q study can be found elsewhere [28]. Individuals were purposively recruited based on demographic, treatment experiences, and other characteristics relevant to the subject of the study [29].

The eligibility criteria for participation were: 16 years of age or older, having received the DSM diagnosis of “Gender Dysphoria”, receiving or having received gender-affirming medical treatment in the Netherlands, sufficient proficiency in Dutch or English, and possessing the ability to provide written and oral informed consent. Participants were recruited by healthcare providers from [blinded for review], or through support groups. Recruitment took place during a three-month period (August 2018–October 2018). Participation was voluntary and had no impact on their present or future gender-affirming treatments. Participants were offered an EUR 50 (euro) gift card for participating, which they were informed about after participation.

In total, 57 transgender individuals were approached, of whom 21 agreed to participate (37%). Due to the mixed sampling strategy, no data of non-responders were collected. Two participants were excluded for the current study because they had not started gender-affirming treatments at time of participation. This resulted in a study cohort of 19 individuals of which 12 identified as (transgender) male, six as (transgender) female and one as transgender.

### 2.3. Data Collection

All interviews were conducted one-on-one and in-person by one member of the research team (M.R.). The interviewer was a non-binary identifying psychologist and trained interviewer with prior experience in qualitative research. Most of the interviews were conducted at the participants’ homes and lasted between 55 min and 156 min (mean = 85). Interviews were conducted using a phenomenological approach seeking to capture participants’ experiences and establish a thorough understanding of their shared meaning. The aim was to develop a framework that would be used to evaluate gender-affirming treatments (GENDER-Q) [28]. The interviewer used a semi-structured interview script informed by clinical expertise and earlier research with open-ended questions to guide the conversation. Topics included a range of physical and psychosocial topics: (e.g., experiences within care, treatment outcomes, recovery, mental health, and body image) [28]. Experienced stressors and adaptation were discussed as part of these topics. Participants could pause or stop the interview at any time and could also choose to withdraw study participation at any time, during or following the completion of the interview. All participants completed the interview in one session and no participant withdrew from participation. None of the interviews were required to be redone, and none of the participants used the option to return the transcriptions for alteration and/or to comment.

### 2.4. Data Extraction and Analysis

All interviews were audio recorded, transcribed verbatim, and translated into English—with all identifiable details removed. Data were analyzed inductively using Braun and Clark’s method for thematic analysis [30]. All transcripts were read and re-read by a researcher (A.O.) for potential codes and themes applicable to the present study objective, after which all codes were verified by a second member of the research team (T.G.). Codes were extracted and imported into ATLAS.ti Mac (Version 9.0.22.0) for analysis.

Analyses of the data were conducted in three rounds: 1/3rd of the data were used in a first round to generate a hypothetical framework with major, minor, and mini themes; in a second round, the proposed framework, codes, and themes (names) were updated using another 1/3rd of the data, and the last 1/3rd of the data were used for internal validation of the framework and to finally adjust the themes and definitions. In all rounds the codes were identified by A.O. and verified by T.G. The researchers discussed the codes and themes until consensus was reached. Ultimately, all findings were subcategorized into stressors, coping strategies and moderating factors. Lastly, illustrative quotes were selected for each theme.

## 3. Results

### 3.1. Sociodemographic Characteristics

The characteristics of the participating cohort are displayed in Table 1. The age of the participants ranged from 18 to 62 years with a mean age of 39 years (SD = 15). All participants received at least hormonal replacement therapy (HRT) and 14 participants also received at least one type of gender-affirming surgery.

### 3.2. General Overview

A total of 1204 codes were extracted, of which 335 were related to stressors and 869 were related to coping strategies. The overall relationship between experienced stressors, coping strategies, identified moderating factors and psychosocial outcomes during/after gender-affirming treatments has been summarized in Figure 1. Strategies used to cope with the identified stressors have been categorized into two distinct groups: adaptive and maladaptive. Classification of strategies as (mal)adaptive was made by the researchers using generalized theories of coping strategies [25,26]. Adaptive strategies were associated with self-assurance, the reduction in physical stress, and affirmative emotions such as happiness, whereas maladaptive strategies were associated with negative outcomes such as psychological distress and depression. Both groups of coping strategies were divided into two subgroups: cognitive strategies and behavioral strategies. Emotional traits were mostly found to be moderating and, therefore, classified along with other moderating factors.

### 3.3. Stressors

The identified stressors and illustrative quotes have been summarized in Table 2. Thematic analysis revealed multiple stressors which could be divided into four major groups: lack of support system (e.g., lack of support from friends or family); transition related stressors (e.g., not being informed enough, dissatisfaction with healthcare providers); physical stressors post-transition (e.g., complications); and psychosocial stressors post-transition (e.g., expectations were not met). While a number of stressors were related to the transition itself, the number of stressors connected to the period following the individual’s gender-affirming medical treatment was also notable.

### 3.4. Moderating Factors

Two groups of factors moderating the relationship between stressors, coping strategies and psychosocial outcomes were identified, namely, experienced support and emotional (coping) traits. Emotional traits which were considered to moderate adaptive coping strategies and improve psychosocial outcomes included resilience, optimism, flexibility (the willingness to change or compromise according to the situation), and openness. Most the participants reporting such traits described themselves as happy and content with the physical and mental changes experienced during/after transition. Optimism and flexibility seemed to help some participants to overcome stressors more easily, and also seemed to help to create a more positive outlook on the future. Openness supported participants in seeking help and prevented social isolation. Participants who showed resilience often reported stamina in overcoming stressors and were quicker to recover from setbacks. With regard to resilience, one participant indicated:

“I went all-in with the transition and accepted everything: the ups and the downs. I just know that, even during the darkest times, you will always climb up again to see the light. Hard times will make you stronger.”—Participant 01, male, age 52.

Additionally, multiple negative emotional traits were identified, including anger, pessimism, and fear. These traits were seen alongside maladaptive coping strategies and poorer psychosocial outcomes such as disappointment and lack of acceptance. Participants who reported pessimism and fear also frequently reported depressive feelings, anxiety, and anger. Individuals who showed a tendency to react with anger, regardless of the stressor, were often less satisfied with the results.

In addition, experiences of support seemed to moderate participants’ tendency to use either adaptive or maladaptive coping strategies. On one side, participants who experienced social support often felt accepted and consequently used more adaptive coping strategies such as self-acceptance and seeking help. In contrast, in the absence of social support participants leaned towards maladaptive strategies such as isolation and lower levels of experienced self-acceptance.

### 3.5. Adaptive Coping Strategies

Six adaptive coping strategies were identified, of which some were subdivided into mini themes. Participants described that using adaptive coping strategies enabled greater well-being and positive emotions during and after transitioning. Two types of coping strategies were identified: cognitive and behavioral strategies. Table 3 shows all adaptive coping strategies, structured by major, minor, and mini themes.

The following three cognitive coping strategies were reported by participants: acceptance; adaptive cognitions concerning gender and transition; and rationalization. The most frequently mentioned adaptive strategy was acceptance (119 quotes). This strategy was used to cope with different stressors, such as accepting one’s (transgender) identity, disappointing results, or one’s appearance. Specifically, participants described acceptance as being helpful in feeling more positive emotions regarding the aforementioned subjects. When participants reported acceptance, for example of one’s appearance, even the parts that were not in line with the individual’s gender identity (ideals), they also reported feelings of satisfaction, perceived control and self-assurance. One participant described the effect of accepting feelings of gender incongruence as:

“Emotionally, I am very stable now. I think that is because I accepted my male sides as well as my female sides.”—Participant 08, female, age 56.

A second, frequently mentioned cognitive strategy included having adaptive cognitions concerning one’s gender and transition. This strategy was used to cope with feelings of incongruence and the individual’s body post- gender-affirming treatment(s). The use of adaptive cognitions meant that participants dismissed stereotypical images of gendered appearance and roles. For many participants this decreased negative emotions that one may have felt if they did not fit perfectly within one of these images.

Rationalization, defined as the effort to explain or justify stressors with logical reasoning [21], was also used to cope with stressors such as a lack of acceptance by friends and family, or social stigmatization. Participants used rationalization as an attempt to understand the environment, as well as themselves (self-knowledge). Rationalization helped to put negative opinions and comments into context and, as a result, helped to relieve emotional distress. One participant describes how rationalizing helps her understand her father’s response to her wish to transition, and thereby helps to accept his point of:

“My dad still asks me why I want to be a girl. But I get it, he is 87 and has called me by my former name my entire life. I get why he still calls me that, it does not bother me. He just doesn’t really get it, that some men want to be women, you know.”—Participant 09, transgender, age 58.

Participants described three types of adaptive coping behavior strategies: seeking help and guidance; taking autonomy during transition; and problem solving. Seeking help and guidance with others was frequently used to cope with feelings of loneliness and isolation and helped to alleviate said feelings. Several participants sought help by actively looking for fellow transgender individuals, seeking out friends or creating an online support system. One participant said, after meeting up with a transgender-support group:

“It was a relief to realize that I am not alone. I’m not insane, I’m not a freak. It is good to know that there are others like me.”—Participant 10, female, age 21.

Additionally, some participants explained that in being autonomous during the transition it enabled the individual to feel more in control and self-assured, especially regarding the transition process. Specifically, individuals described taking non-medical steps to support transition, including physical (e.g., wearing chest binders) or social steps (e.g., dressing in line with the identified gender when going out). While also experimenting with gender and feeling more in control in the process, participants described that it was such behavior that enabled them to start transitioning while waiting for medical care. For example, one participant described that using non-medical interventions made her feel more gender-congruent:

“I made a tight band with a sock so I could bind the penis tightly to the back. When you wear pants, you cannot see a bump. I felt more like myself.”—Participant 11, female, age 28.

Lastly, participants described, generally, that problem-solving skills helped them to cope with stressors throughout the transitioning process. By actively problem solving, stressors could be resolved before they became intolerable, and thus, negative emotions or repercussions could be prevented. Sometimes problem solving was used to prevent undesirable situations from happening (e.g., by actively approaching people). Some participants described taking the initiative to explain their situation or feelings which, as a result, led them to feel more self-assured and understood. One participant discussed how he and his partner talked about the ways in which they could have intercourse in a way they both enjoyed:

“My partner and I discussed it beforehand, so there would be no unpleasant surprises. We talked about what she would like and what I would like and what we could do to make it pleasant for both of us. We talked about it for a long time until we both felt good about it.”—Participant 12, male/transgender male, age 18.

### 3.6. Maladaptive Coping Strategies

A total of seven maladaptive coping strategies were identified in response to experienced stressors, of which some were subdivided into mini themes. Table 4 shows all maladaptive coping strategies.

In contrast to adaptive coping, maladaptive coping strategies often led to negative emotions, decreased self-worth and less favorable future perspectives. Like the adaptive strategies, cognitive and behavioral coping styles were identified.

The cognitive maladaptive coping styles were subdivided into four main strategies: lack of self-acceptance, maladaptive cognitions concerning gender and transition, external validation of self-esteem, and externalization. It was common for participants to talk about the difficulty they had with accepting themselves, which led to stress, and negative or depressive feelings. Some struggled to accept being transgender, which, at times was due to internalized transphobia or shame. Other participants had difficulty accepting the gender non-conforming parts of their appearance as they wanted to fit into a stereotypical gendered ideal. One participant talked about his lack of self-acceptance:

“I can hardly look at myself in the mirror. When I’m naked, my confidence is almost zero.”—Participant 12, male/transgender male, age 18.

Participants frequently mentioned having maladaptive cognitions concerning gender and transition, which included having stereotypical images of men/women, focusing on gender-incongruent characteristics and experiencing internalized transphobia. When participants were mainly focused on gender-incongruent characteristics they were not easily satisfied with the results of the treatments, leading to increased chances of reporting negative emotions, discomfort and/or requesting re-operations and surgical corrections. This mostly applied to (but was not necessarily limited to) gendered body characteristics such as breasts, curves of the body and facial hair. When participants focused especially on gendered characteristics or had high expectations, extra operations or treatments were often sought. In some, gendered physical ideals were stereotypical and the variation in physical characteristics within cisgender individuals was often dismissed. One participant said:

“I was not very happy with the results, that was difficult. The form and outline of my face was too feminine. I still had boob tissue, and the skin was loose. So, I went back to have another operation to make it look more masculine.”—Participant 07, male/transgender male, age 27.

Thoughts that reflect internalized transphobia were also repeatedly observed. Some participants described internalized negative attitudes towards other transgender individuals in general, and/or towards being transgender themselves. This included internalized disapproval towards transgender individuals, uneasiness with disclosing one’s identity to others and discomfort with being compared to other transgender individuals. Internalized transphobia sometimes led to internal conflicts, lowered levels of self-respect, difficulty accepting being transgender and sometimes depressive feelings. One participant indicated:

“I feel very negatively towards being transgender. I am always afraid that other people notice it and talk about me. If I could make one wish, I would wish that I was not transgender. Then I wouldn’t have all the problems in my life, and I would be able to live normally.”—Participant 11, female, age 28.

External validation of self-esteem was sometimes reported in relation to feeling self-assured, more specifically in the validation of gender-related esteem. In this case, confirmation was sought in friends, family, and strangers to feel assured about gender-typical characteristics, and made participants feel more self-assured. However, when not validated, participants felt more self-conscious and anxious. For example, one participant described her levels of self-esteem and consciousness as being dependent on being correctly or incorrectly gendered:

“I am always aware of other people’s reaction to me. How they look at me and what they think of me. […] The other day, I heard someone ask: is that a man or a woman? That bothered me a lot. Immediately, I wondered what I had done to be viewed masculine. Did I walk too fast or look irritated or behave odd?”—Participant 03, female, age 52.

The last cognitive strategy identified was externalization. Some participants used this strategy to alleviate the emotional pressure of feeling different by attributing negative situations to others. This happened most often when participants still struggled with (gendered) norms and found it difficult to get past these norms:

“Sometimes I think that I’m not the problem, the rest of the world is. They are all confused because I am confused about my gender. But who is really the problem? Not me!”—Participant 13, male, age 40.

Participants indicated the use of several types of maladaptive behaviors to cope with stressors. Three behavioral strategies were identified: isolation, avoidance, and self-destructive behavior. It was common for participants to socially isolate themselves for longer or shorter periods, to avoid confrontation and possible rejection. A variety of isolation methods were described, including social isolation, emotional isolation, and non-communication. Isolation could lead to feelings of rejection and loneliness and increased the risk of depressive feelings. One participant explained:

“Isolation is the real problem when you grow up with gender incongruence. You can never live up to the expectations, so you always feel like you must hide a part of yourself. You feel like you are alone in the world. I still isolate myself when I feel bad.”—Participant 14, male/transgender male, age 62.

In addition, avoidance, in any form, was frequently described. To some, avoidant behavior was an unconscious effort to alleviate feelings of being different and feeling rejected. Frequently, social situations were avoided in order to evade possible negative reactions or stigmatization. In general, avoidance and the unwillingness to seek and accept help when appropriate often led to maintaining negative feelings and behaviors towards both oneself and the other, thereby making participants more vulnerable to a decrease in psychosocial wellbeing. One participant stated:

“Transgender individuals do not often ask for help. We are just used to doing everything alone, that’s a hard habit to break. I always tried to solve everything by myself.”—Participant 14, male/transgender male, age 62.

Finally, self-destructive behavior was used to avoid negative emotions that were linked to the former gender identity. Feelings of low self-esteem or even self-loathing could trigger self-destructive behaviors such as alcohol and/or substance abuse. One participant talks about his substance abuse:

“I was stoned for a very long time, almost 20 years. I tried to dull a lot of pain and suffering by smoking weed.”—Participant 13, male, age 40.

## 4. Discussion

The aim of this pilot study was to identify what stressors were commonly experienced during and after gender-affirming treatments, and to then gain further understanding into the resulting coping strategies that transgender individuals use. While studies have shown that gender-affirming treatments improve the quality of life for transgender individuals that seek medical transition, little is known about how these individuals adapt to stressors during and after transition [12]. Our findings indicate that transgender individuals may experience a range of stressors after this period and use several coping strategies to adapt. Consistent with findings in cisgender populations, this explorative study confirmed two types of coping strategies: adaptive and maladaptive coping, which were further subdivided into cognitive and behavioral strategies, which were again moderated by experienced support and emotional traits [25]. Both stressors and coping strategies included transgender-specific as well as non-specific characteristics.

### 4.1. Most-Reported Coping Strategies

In our explorative study, cognitive strategies were the most-reported adaptive coping styles. Acceptance was often central in adaptive coping and related to both acceptance of social experiences and of oneself. It was suggested that experiencing greater acceptance provided higher levels of perceived control, encouraged positive emotions, and was associated with reporting other adaptive coping strategies such as help-seeking, problem solving and having adaptive cognitions regarding gender/transition as well. Similar results on the role of acceptant coping were found in cisgender populations with a chronic illness [31,32,33]. Higher levels of acceptance towards chronic illnesses are generally related to fewer negative emotions and to practicing adequate problem-solving approaches and other adaptive strategies [31,32,33]. Although recognizing that gender dysphoria is not a chronic illness, and experiences of transgender individuals may differ greatly from those with chronic illnesses, both groups may be confronted with deviating from societal norms, going through significant changes in life and/or having to deal with physical impairments as a result of (complicated) medical treatments.

While cognitive strategies were the most-reported adaptive coping styles, more variation was reported in the maladaptive strategies. Our study found isolation to be the most common maladaptive behavioral strategy, a strategy that was mostly used to avoid confrontation, stigmatization, and rejection. Participants noted a variety of isolation methods including social isolation, emotional isolation, and non-communication, which was associated with feelings of loneliness and risk of depression. Social isolation has been reported alongside feelings of not-belonging, failure to engage with others and neglect of relationships, and subsequently has a negative impact on mental health [34,35]. Available data on lesbian, gay and bisexual (LGB) individuals show similar results [36]: isolation is a frequently mentioned strategy by LGB individuals to hide one’s identity in fear of discrimination and rejection, yet it increases the risk of developing psychopathology [37]. Both LGB and transgender individuals may experience social stressors, as well as internalized rejection and decreased self-worth which one may try to avoid by isolating.

### 4.2. Moderating Factors

In addition to frequently used coping strategies, our study also identified factors that appeared to moderate the use of adaptive or maladaptive coping strategies. We found that positive emotional traits, such as optimism and resilience, were generally reported in combination with more adaptive coping behaviors and cognitions. This finding is in line with available data on lesbian, gay, bisexual, transgender, and queer (LGBTQ) youth [38]. High levels of optimism and resilience appear associated with a tendency to expect more positive results in the future and are associated with adaptive coping styles, therefore, contributing positively to psychosocial wellbeing [38]. Research into cisgender patients with breast cancer similarly showed that positive emotional traits improved the quality of life, enhanced physical health and decreased distress [39]. Although referring to different populations, both groups may go through extensive medical treatments and/or deal with norms on gendered physical appearance.

In contrast, negative emotional traits such as anger and pessimism were expressed in combination with more maladaptive coping behaviors and cognitions and appeared to be linked with feeling less satisfied about the individual’s results. Studies show that individuals who exhibit negative emotional traits can respond inadequately to environmental stress, experience minor stressors as overwhelming, and have elevated levels of anxiety and depression [37,40]. This can also lead to the additional adoption of maladaptive coping strategies [25].

Furthermore, this study highlighted that social support is an important moderating factor in the individual’s tendency to use either adaptive or maladaptive strategies. Individuals who reported more social support also reported adaptive coping strategies more frequently, and vice versa. In general, multiple studies have shown that social support can serve as a moderating factor in preventing the development of mental health issues in cisgender individuals [41], LGB individuals, [42,43] and transgender individuals [27]. Support can come from many sources (e.g., friends, family, transgender peers, or healthcare providers) and is thought to be important for stimulating active engagement, preventing isolation, and promoting self-acceptance [42,43].

### 4.3. General and Transgender Specific Coping Strategies

The current pilot study confirmed a range of general coping strategies that previously had been described in cisgender populations [25], but also identified additional coping strategies specific for transgender individuals and the stressors that they experience.

Our study identified coping strategies similar to those recognized in (cisgender) individuals who have to manage major life events or (chronic) illness, such as breast cancer [44,45]. Adaptive strategies, such as acceptance and rationalization, are often reported in these groups [44,45]. Furthermore, coping strategies that mitigate similar kinds of social stress, like that which occurs during sexual identity formation in LGB individuals, were also verified for transgender individuals [25,46]. This included adaptive strategies such as seeking help and problem solving, and maladaptive strategies such as lack of acceptance, isolation, and self-destructive behavior [47,48,49].

The transgender specific strategies we identified were mostly related to (mal)adaptive cognitions concerning gender (identity or presentation) and transition. Maladaptive cognitions included internalized transphobia or stereotypical (binary) images of gender and appearance. In contrast, adaptive cognitions included non-stereotypical inclusive images. These specific strategies are the result of specific stressors that transgender individuals are exposed to during and after transition, such as stigmatization, gendered norms (e.g., attitudes, appearance, or behavior), and victimization which can cause significant psychological distress [19,20,50,51]. Moreover, a longing to pass and conform to these gendered norms can introduce additional stress to some individuals [20,50,51]. Previous studies have found the damaging effects that gender-related stigma and stressors can have on the mental health of transgender individuals [20,51,52]. Specific coping mechanisms such as internalized transphobia and stereotypical image of gender norms may be adopted in response to gender-related stigma/stressors [53]. However, this was also reported in other minority populations, such as individuals who experience racism or discrimination [54,55]. Further research is needed to identify who will develop (mal)adaptive coping in response to transgender-specific stressors and on possible interventions within this group.

### 4.4. Possible Approaches for Intervention

Both the observed stressors, as well as the (mal)adaptive coping styles, used by transgender individuals after gender-affirming treatments may provide possible approaches for interventions that could improve the mental health of transgender individuals seeking to medically transition. Regarding the sources of stressors, a substantial number of participants reported stressors experienced (in)directly from the process of gender-affirming care. Stressors included dissatisfaction with the amount, and content, of information given pre-treatment, in addition to disappointment with overall results. Participants who underwent genital gender-affirming surgeries reported this stressor especially. For example, after the penile inversion vaginoplasty in transgender women, lifelong interventions such as frequent douching and dilatation of the neovagina are required to maintain its function [14,56]. With adequate counseling on these long-term requirements, in addition to examples of functional and aesthetic outcomes, the individual’s post-treatment distress and dissatisfaction could be reduced [57]. These findings regarding the importance of adequate pre-treatment counseling have also been linked to improved recovery experiences in cisgender populations before [58].

In addition to the prevention of healthcare-associated stressors, some frequently reported coping strategies could actually provide an avenue for possible intervention. Adaptive strategies such as acceptance and seeking help could be promoted, while the negative effects of maladaptive strategies such as lack of self-acceptance and isolation could be identified, and subsequently individuals could be made aware of its negative effects. Interventions aiming to promote adaptive coping strategies have a proven positive effect on mental health outcomes [59,60]. Austin and Craig have developed the AFFIRM module, a cognitive behavioral intervention developed to improve adaptive coping and decrease mental health problems for sexual and gender minority youth [61]. AFFIRM provides clients with understanding of cognitions, mood, and behavior and gives guidance on how to identify and use skills to optimize coping strategies [61]. Studies in LGBTQ youth showed that the use of AFFIRM increased the engagement of adaptive coping mechanisms such as problem solving and seeking help, promoted resilience, and reduced maladaptive coping strategies [61,62]. Similarly, multiple studies on LGBTQ and non-LGB cisgender individuals found that cognitive behavioral therapy, in general, has a positive effect on psychosocial symptoms, promoted adaptive coping strategies as problem solving and resilience, as well as reduced overall use of maladaptive strategies [62]. A recent study also highlighted the need for psychological aftercare following gender-affirming treatments [14]. Further development of gender-affirming healthcare services can therefore include coping–supportive (after)care as part of gender-affirming medical treatment.

### 4.5. Limitations

Our pilot study was subject to several limitations. Since secondary analyses were performed on existing data, participants may have experienced additional stressors and used coping mechanisms that were not revealed by the present research method. However, spontaneously discussed coping strategies and stressors were thought to reflect the main coping mechanisms of the participants. Moreover, with at least 10–20 participant codes for most coping strategies, sufficient saturation for the reported coping styles can be assumed. However, in future research an existing measuring tool to capture coping styles could be used/validated for this population (e.g., A-COPE) [63]. Additionally, since secondary analyses were performed on existing interview data, the researchers classified coping strategies as adaptive or maladaptive, rather than the participants themselves. Therefore, future research should discuss how the participants experience different (mal)adaptive coping strategies themselves. Some strategies that are categorized as maladaptive may sometimes be adaptive as well, a concept echoed within cisgender literature [25,26]. This can be achieved by qualitative research using interviews or by quantitative research using existing tools to capture coping styles. Furthermore, the moderating factors were identified exploratively by the researchers, but no formal moderator analysis was performed. Therefore, assumptions based on the moderating factors and their effects should be made with caution. Furthermore, given the relatively small sample size, and the wide range of age and treatment paths, results may not be representative for transgender individuals with various sociodemographic and clinical characteristics. In future research, sociodemographic and clinical characteristics could be associated with specific stressors/experiences and coping styles. Participants were recruited through the [blinded for review] and support groups. Thereby it is possible that participants were more open and actively interested in transgender matters when compared to transgender individuals in the general population. Henceforth, assumptions and generalizations based on the current study should be made cautiously. Future research to validate the findings of this explorative study should use a larger and more generalizable population. Moreover, participants were evaluated cross-sectionally, thus it may be beneficial for further research to follow transgender individuals throughout their transition, taking various points of time into consideration both during and following transition to evaluate possible effects. Furthermore, not many data are available on coping strategies used by transgender individuals; therefore we compared our results to studies on patients with chronic illness instead. We recognize that gender dysphoria is no illness, and therefore, experiences may differ in both groups. Since the current research was a pilot study, some factors were not fully explored, such as the impact of specific gender-affirming treatments. Lastly, the purpose of this study was to describe coping and stressors during and after a variety of gender-affirming procedures, yet within the interviewing process some impactful pre-treatment experiences were also mentioned. Therefore, a possible exploration for future research could be the full relationship between pre-treatment experiences and coping strategies, as well as coping post-transition.

The present explorative study provided insight into different stressors that transgender individuals may be exposed to during and following gender-affirming treatments, as well as demonstrated the variation in coping strategies that individuals used when dealing with these stressors. The stressors related to the gender-affirming medical treatments and social experiences that occurred during this period. In order to enhance long-term positive mental health outcomes, being aware of frequently occurring stressors, understanding, and subsequently addressing the different types of coping strategies that transgender individuals use may be helpful. Further research with a more specific research design focused on coping strategies and targeted interview questions is needed to differentiate between general and transgender (stressor) specific coping strategies. In addition to identifying these coping strategies, future research might also look more explicitly at possible approaches for therapy.

## Figures and Tables

**Figure 1 healthcare-11-00089-f001:**
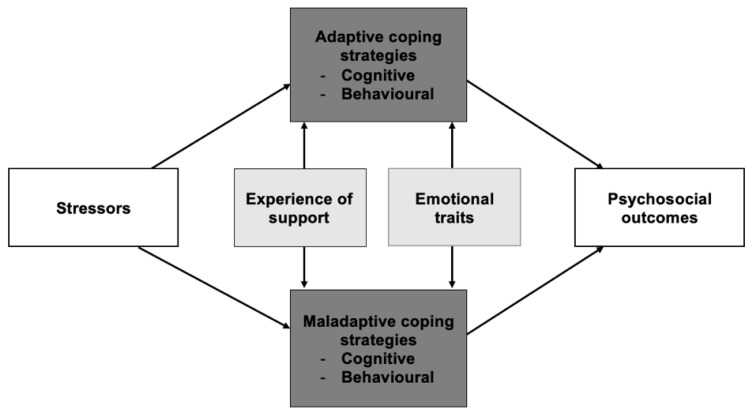
Model of stressors, coping strategies, psychosocial outcomes, and moderating factors.

**Table 1 healthcare-11-00089-t001:** Participant characteristics.

ID	Age	Gender Identity	Sex Assigned at Birth	Gender-Affirming Treatments
1	52	Male	Female	HRT
2	26	Male/Transgender male	Female	HRT, mastectomy
3	52	Female	Male	HRT, vaginoplasty, chondrolaryngoplasty
4	41	Male	Female	HRT, mastectomy, hysterectomy, bilateral salpingo-oophorectomy, colpectomy, metoidioplasty
5	60	Male	Female	HRT, mastectomy, hysterectomy, bilateral salpingo-oophorectomy, metoidioplasty with urethral lengthening, phalloplasty with urethral lengthening, erectile prosthetic, testicle implants
6	23	Female	Male	HRT, puberty inhibitors
7	27	Male/Transgender male	Female	HRT, mastectomy, hysterectomy, bilateral salpingo-oophorectomy, colpectomy
8	56	Female	Male	HRT, vaginoplasty
9	58	Transgender	Male	HRT
10	21	Female	Male	HRT, puberty inhibitors, orchidectomy
11	28	Female	Male	HRT
12	18	Male/Transgender male	Female	HRT
13	40	Male	Female	HRT, mastectomy, hysterectomy
14	62	Male/Transgender male	Female	HRT, mastectomy, hysterectomy, bilateral salpingo-oophorectomy, metoidioplasty
15	26	Male/Transgender male	Female	HRT, mastectomy
16	27	Male	Female	HRT, mastectomy
17	22	Male	Female	HRT, mastectomy, hysterectomy, bilateral salpingo-oophorectomy
18	61	Female	Male	HRT, breast augmentation, FFS, chondrolaryngoplasty, vaginoplasty, glottoplasty
19	37	Male	Female	HRT, mastectomy, hysterectomy, colpectomy, bilateral salpingo-oophorectomy

HRT: Hormonal Replacement Therapy, FFS: Facial Feminization Surgery.

**Table 2 healthcare-11-00089-t002:** Stressors reported during and after gender-affirming treatment.

Stressors			
Major Theme	Minor Theme	N=	Quote (Participant Number)
Lack of support system	Lack of acceptance or support from family or friends	47	When we were at the zoo, I told my mom that I would like her to call me [new name], but she shouted at me: I will never ever accept you! (01)
Transition related	Not being informed enough during medical transition	62	I was carrying this weight with me until I finally spoke with the surgeon. I thought: why didn’t you tell me this EARLIER? I would have been just fine knowing, but I would have liked to know. (02)
	Dissatisfaction with healthcare providers during medical transition	48	I often feel like the doctor does not listen to me, that is very disappointing. (03)
	Missing guidance with social transition	31	I would advise to focus more on guidance when you start the transition. What does it mean? How do you present yourself? How do you handle people’s reactions when they don’t know how to address you? I would have liked that kind of guidance. When am I going to do the social transition? How am I going to approach that? (04)
	Revision surgery	11	When I got the protheses, it did not fit properly. I had to return to the hospital several times. In total, I had to undergo 14–15 operations to fix the problem. (05)
Post-transition: physical	Complications	33	My labia minora were turning black at some point because they were dying. (03)
	Diminished sexual pleasure	13	Not good, my libido has decreased enormously since I started the hormonal replacement therapy. (06)
Post-transition: psychosocial	Doubts about transition	24	Sometimes I feel a little schizophrenic or something. There’s a part of me, the biggest part, that feels like a man but there’s another part that’s not comfortable with it and it is a constant internal conflict. (07)
	Had to get used to the physical and mental changes	23	Before the transition, I could easily carry heavy bags and stuff, but now I struggle to do so. My strength has decreased a lot. Last week, I wanted to carry a suitcase for a friend, but I realized that I could no longer do so. Somebody else had to carry it. (08)
	Transition does not solve all problems	23	I hoped or thought that transitioning would make me happy. At the very beginning you think: if I transition then I am happy, but it doesn’t turn out that way. It does help, but it is not going to be the sole reason for happiness. (07)
	Expectations were not met	20	Some things you don’t expect. And of course, everyone experiences that in their own way. Some results turned out differently than I expected, but that does not mean it went wrong, probably my expectations were wrong. (05)

N = refers to number of quotes.

**Table 3 healthcare-11-00089-t003:** Adaptive coping strategies.

Adaptive Coping Strategies			
Major Theme	Minor Theme	Mini Theme	N=
Cognitive	Acceptance		119
	Adaptive cognitions concerning gender and transition	Not thinking binary	32
	Rationalizing	Self-knowledge	51
Behavioural	Seeking help and guidance	Seeking help/support	44
		Finding (spiritual) meaning	21
	Autonomy in arranging transition	Taking small steps	21
		Non-medical interventions	26
	Problem-solving	Confronting	24

N = refers to number of quotes.

**Table 4 healthcare-11-00089-t004:** Maladaptive coping strategies.

Maladaptive Coping Strategies			
Major Theme	Minor Theme	Mini Theme	N=
Cognitive	Lack of self-acceptance		36
	Maladaptive cognitions concerning gender and transition	Stereotypical image of man/woman	75
		Focus on gender-incongruent characteristics	66
		Internalized transphobia	39
	External validation of self-esteem	Comparing to others	53
	Externalization		21
Behavioural	Isolation	Denial of identity	87
		Hiding	70
	Avoidance	Solving alone	32
		Dissociation	33
	Self-destructive behaviour	Substance abuse	19

N = refers to number of quotes.

## Data Availability

The data presented in this study is not publicly available due to privacy restrictions.

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
