# Peer review of "Identifying Coping Strategies Used by Transgender Individuals in Response to Stressors during and after Gender-Affirming Treatments—An Explorative Study"

_healthcare, 2022, doi:10.3390/healthcare11010089_

Round 1

Reviewer 1 Report

In this study, the authors have undertaken qualitative interviews to better understand the treatment outcomes and health care experiences of Dutch transgender people who have received gender-affirming procedures. The study uses data from 12 transgender males, six transgender females, and one transgender person. Using inductive coding and theory-informed theme analysis, the authors assessed four stressor domains—lack of a support structure, stressors connected to transition, and physical and mental stressors post-transition. Further, the authors identified acceptance, help seeking, and adaptive cognitions about gender and transition as the most often reported adaptive coping mechanisms, while social isolation and maladaptive cognitions about gender and transition were identified as maladaptive coping mechanisms. Together, these findings suggest that transgender people may endure severe stress before and after gender-affirming therapies and, as a result, employ a wide range of coping techniques, both transgender-specific and non-specific.

This work does, in my opinion, help us get a better idea of how stress and (mal)adaptive ways of coping can help improve mental health care and support for LGBT people. It does this by giving us a better idea of the different kinds of stressors that transgender people face before and after gender-affirming therapies, as well as the different ways they use to deal with them.

However, as the authors recognized, there are limitations to generalizations. I think the authors should come up with more specific ways for other researchers with similar research interests and for their own future work to use the theme and results of this study. 

Author Response

In this study, the authors have undertaken qualitative interviews to better understand the treatment outcomes and health care experiences of Dutch transgender people who have received gender-affirming procedures. The study uses data from 12 transgender males, six transgender females, and one transgender person. Using inductive coding and theory-informed theme analysis, the authors assessed four stressor domains—lack of a support structure, stressors connected to transition, and physical and mental stressors post-transition. Further, the authors identified acceptance, help seeking, and adaptive cognitions about gender and transition as the most often reported adaptive coping mechanisms, while social isolation and maladaptive cognitions about gender and transition were identified as maladaptive coping mechanisms. Together, these findings suggest that transgender people may endure severe stress before and after gender-affirming therapies and, as a result, employ a wide range of coping techniques, both transgender-specific and non-specific.

This work does, in my opinion, help us get a better idea of how stress and (mal)adaptive ways of coping can help improve mental health care and support for LGBT people. It does this by giving us a better idea of the different kinds of stressors that transgender people face before and after gender-affirming therapies, as well as the different ways they use to deal with them.

Response:

We would like to thank reviewer 1 for the constructive and positive comments on our manuscript.

However, as the authors recognized, there are limitations to generalizations. I think the authors should come up with more specific ways for other researchers with similar research interests and for their own future work to use the theme and results of this study. 

Response:

We agree with this comment, and we incorporated it in the limitations:

“Additionally, since secondary analyses were performed on existing interview data, the researchers classified coping strategies as adaptive or maladaptive, rather than the participants themselves. Therefore, future research should discuss how the participants experience different (mal)adaptive coping strategies themselves. Some strategies that are categorized as maladaptive may sometimes be adaptive as well, a concept echoed within cisgender literature ((25, 26). This can be achieved by qualitative research using interviews or by quantitative research using existing tools to capture coping styles.” (Page 14 line 1035-1042).

Reviewer 2 Report

Thank you to the authors for their hard work and careful attention to detail in conducting the study described here. As a biostatistician with formal training in study design, qualitative as well as quantitative methods, and observational studies, I found this study interesting, informative, and fascinating. Nonetheless, there remain some significant limitations to the paper that will need to be addressed in subsequent versions to make it appealing to readers in the healthcare literature. I could not find the tables referred to in the paper, which will work against this review. 

I found this paper to be well-written and well-composed with respect to technical writing. Except for the lack of tables and figures referred to in the text, the sections were well-composed with all of the relevant parts. Sentences and paragraphs flowed nicely throughout the paper. The text was easy to follow and telegraphed nicely the forthcoming information within and across sections. Statements used to support points made by the study team were very often on-point and rich in their detail. Most importantly, the study team has taken an extremely important and difficult issue within the medical community and written about it with sensitivity, depth, and care. Their work identifies directly many of the nuances undergirding the concepts, belief systems, and thought patterns of the patient sample.   

Less positive, however, are some characteristics that seem to detract from the overall integrity of the paper. At a basic level, the term ‘data’ is actually plural rather than singular. In addition, the keyword "bioethid" is not familiar to me. These points should be corrected in subsequent versions of the paper. In addition, my sense is that the paper was much longer than it needed to be with an excessive number of redundant and/or less informative points, both in concept and expression of details. Defining rather simple scientific terms as stress (pg 2), coping strategies (pg 2), and rationalization (pg 6) seem excessive and unnecessary. This would suggest that the outlet may not be the correct one for the intended audience. Some sections were beautifully written and fully detailed (i.e., maladaptive coping strategies) while others were not (i.e., moderating factors, adaptive strategies). In the case of the former, the themes and subthemes appeared to be identified and well-supported throughout the entire section with a coherence that linked each one to the others and generalizations that flowed easily from the identified points/statements (e.g., conditions, validation, avoidance; pgs 7-8). In contrast, I did not detect support for the concept of a moderator (pg 5). My sense is that this term carries with it specific interpretations, at least in the scientific literature, that were glossed over.  Further, there seemed to be statements made that reached beyond the strength of the information obtained in the interviews (e.g., experiences moderated the use of adaptive/maladaptive strategies).  Ironically, and despite my objection to defining earlier terms, this term could have benefited from defining with greater support for the interpretations. For adaptive strategies, specifically, it seemed as if the evidence in support of adaptive cognition and rationalization (pg 5-6) could have been stronger and more tightly linked to the statements provided. The authors may wish to include multiple statements to support many of their stronger and more prominent points—to strengthen their case. Finally, and importantly, the conclusion reached and reported in the Abstract seemed unremarkable: transgender individuals may undergo significant stress during and after gender affirming treatments… the authors may wish to expand upon this finding and dive deeper into the specifics of their overall conclusion.  As currently written, this conclusion could be said about any observational report of a life-altering intervention. While it may be true, their study undoubtedly goes beyond this very basic point. 

 In conclusion, I found this paper to be interesting, informative, and authentic in its detail. The authors have done a nice job of taking on a supremely difficult healthcare topic and studying it with care and attention. The writing was clear and the paper well-composed.  However, the paper had several weaknesses that detracted from the overall integrity of the reporting and write up of their study, including excessive explanation of many of their more obvious points, unbalanced development and support of specific themes and subthemes, and a conclusion that appeared generic and lacked specificity to this particular pilot study. The fact that the supporting tables were not available at the time of this review served as a major limitation to the review. In is current form, my sense is that the paper is too early in development to be of interest to the Healthcare readership at this time.       

Author Response

Thank you to the authors for their hard work and careful attention to detail in conducting the study described here. As a biostatistician with formal training in study design, qualitative as well as quantitative methods, and observational studies, I found this study interesting, informative, and fascinating. Nonetheless, there remain some significant limitations to the paper that will need to be addressed in subsequent versions to make it appealing to readers in the healthcare literature. I could not find the tables referred to in the paper, which will work against this review.  

Response:

We would like to thank reviewer 2 for the extensive and constructive feedback. We added the tables and figure to the manuscript.

I found this paper to be well-written and well-composed with respect to technical writing. Except for the lack of tables and figures referred to in the text, the sections were well-composed with all of the relevant parts. Sentences and paragraphs flowed nicely throughout the paper. The text was easy to follow and telegraphed nicely the forthcoming information within and across sections. Statements used to support points made by the study team were very often on-point and rich in their detail. Most importantly, the study team has taken an extremely important and difficult issue within the medical community and written about it with sensitivity, depth, and care. Their work identifies directly many of the nuances undergirding the concepts, belief systems, and thought patterns of the patient sample.   

Less positive, however, are some characteristics that seem to detract from the overall integrity of the paper. At a basic level, the term ‘data’ is actually plural rather than singular.

In addition, the keyword "bioethid" is not familiar to me. These points should be corrected in subsequent versions of the paper.

Response:

Thank you for your feedback. We changed ‘data’ into plural. According to our knowledge, the keyword ‘bioethid’ is not part of our paper.

In addition, my sense is that the paper was much longer than it needed to be with an excessive number of redundant and/or less informative points, both in concept and expression of details. Defining rather simple scientific terms as stress (pg 2), coping strategies (pg 2), and rationalization (pg 6) seem excessive and unnecessary. This would suggest that the outlet may not be the correct one for the intended audience.

Response:

We thank you for your suggestion. Since these simple scientific terms are often defined in different ways and contexts, we chose to keep the definition in our manuscript in order to make sure the definition we used was clear.

Some sections were beautifully written and fully detailed (i.e., maladaptive coping strategies) while others were not (i.e., moderating factors, adaptive strategies).

Response:

The whole manuscript has been revised for language and grammar.

In the case of the former, the themes and subthemes appeared to be identified and well-supported throughout the entire section with a coherence that linked each one to the others and generalizations that flowed easily from the identified points/statements (e.g., conditions, validation, avoidance; pgs 7-8). In contrast, I did not detect support for the concept of a moderator (pg 5). My sense is that this term carries with it specific interpretations, at least in the scientific literature, that were glossed over. 

Response:

We agree with the reviewer’s suggestion, and we added a paragraph in the limitations explaining that the moderating factors were of an explorative nature and no formal moderator analysis was performed:

 “Furthermore, the moderating factors were identified exploratively by the researchers, but no formal moderator analysis was performed. Therefore, assumptions based on the moderating factors and their effects should be made with caution.” (Page 14 line 1042-1045).

Further, there seemed to be statements made that reached beyond the strength of the information obtained in the interviews (e.g., experiences moderated the use of adaptive/maladaptive strategies).  Ironically, and despite my objection to defining earlier terms, this term could have benefited from defining with greater support for the interpretations.

Response:

Thank you for your observations. We have rewritten some paragraphs, including the one on moderating factors and adaptive coping strategies:

Moderating factors, (page 7, line 492-521)

Adaptive coping strategies, (page 7-8, line 522-577).

For adaptive strategies, specifically, it seemed as if the evidence in support of adaptive cognition and rationalization (pg 5-6) could have been stronger and more tightly linked to the statements provided. The authors may wish to include multiple statements to support many of their stronger and more prominent points—to strengthen their case.

Response:

See response above.

Finally, and importantly, the conclusion reached and reported in the Abstract seemed unremarkable: transgender individuals may undergo significant stress during and after gender affirming treatments… the authors may wish to expand upon this finding and dive deeper into the specifics of their overall conclusion.  As currently written, this conclusion could be said about any observational report of a life-altering intervention. While it may be true, their study undoubtedly goes beyond this very basic point. 

Response:

Thank you for your suggestion. We made the conclusion in the abstract and in the discussion more specific by adding the nature of some of the stressors:

Abstract:

“Results of this pilot study showed that transgender individuals may undergo significant stress during and after gender-affirming medical treatment related to the treatments and the social experiences that occur during this period, and as a result, use a range of coping strategies to adapt to the stress.” (Page 1, line 38-40).

Discussion:

“The present explorative study provided insight into different stressors that transgender individuals may be exposed to during and following gender-affirming treatments, as well as demonstrated the variation of coping strategies that individuals used when dealing with these stressors. The stressors related to the gender-affirming medical treatments and social experiences that occurred during this period.” (Page 15, line 1137-1141).

In conclusion, I found this paper to be interesting, informative, and authentic in its detail. The authors have done a nice job of taking on a supremely difficult healthcare topic and studying it with care and attention. The writing was clear and the paper well-composed.  However, the paper had several weaknesses that detracted from the overall integrity of the reporting and write up of their study, including excessive explanation of many of their more obvious points, unbalanced development and support of specific themes and subthemes, and a conclusion that appeared generic and lacked specificity to this particular pilot study. The fact that the supporting tables were not available at the time of this review served as a major limitation to the review. In is current form, my sense is that the paper is too early in development to be of interest to the Healthcare readership at this time.       

Reviewer 3 Report

I consider this a very valuable and important study. And although it is targeted at a specific group of persons (transgender persons having gender-affirming treatment), the study and explanation show that the findings have implications for a broader spectrum as well. The sample size was small, which might be a limitation - as mentioned in the text - but I don't consider this a limitation for publication, it only shows that this is a significant first study with follow-up studies necessary. The only instance where I hesitated was when comparing experiences of transgender persons with the group of individuals suffering from chronic illnesses, "both groups may be confronted with deviating from societal norms". Linking gender-affirming treatments to illnesses (there is also the comparison to breast cancer patients) falls back on notions of labeling gender dysphoria as illness. While I can see the wish for comparing different groups and their coping strategies, this still leaves an impression of transgender individuals being 'sick'. I wish there could be a way of circumventing such analogies, but maybe I am being overly sensitive here. Otherwise this is an admirable study and a well-written elaboration. 

Author Response

I consider this a very valuable and important study. And although it is targeted at a specific group of persons (transgender persons having gender-affirming treatment), the study and explanation show that the findings have implications for a broader spectrum as well. The sample size was small, which might be a limitation - as mentioned in the text - but I don't consider this a limitation for publication, it only shows that this is a significant first study with follow-up studies necessary.

Response:

We would like to thank reviewer 3 for the positive and constructive feedback.

The only instance where I hesitated was when comparing experiences of transgender persons with the group of individuals suffering from chronic illnesses, "both groups may be confronted with deviating from societal norms". Linking gender-affirming treatments to illnesses (there is also the comparison to breast cancer patients) falls back on notions of labeling gender dysphoria as illness. While I can see the wish for comparing different groups and their coping strategies, this still leaves an impression of transgender individuals being 'sick'. I wish there could be a way of circumventing such analogies, but maybe I am being overly sensitive here. Otherwise this is an admirable study and a well-written elaboration. 

Response:

We agree with this comment and have added the following lines in the discussion and limitations:

Discussion:

“Although recognizing that gender dysphoria is not a chronic illness, and experiences of transgender individuals may differ greatly from those with chronic illnesses, both groups may be confronted with deviating from societal norms, going through significant changes in life and/or having to deal with physical impairments as a result of (complicated) medical treatments.” (Page 12, line 856-860).

Limitations:
“Furthermore, not much data is available on coping strategies used by transgender individuals, therefore we compared our results to studies on patients with chronic illness instead. We recognize that gender dysphoria is no illness, and therefore, experiences may differ in both groups.” (Page 15, line 1127-1130).